# Host species is linked to pathogen genotype for the amphibian chytrid fungus (*Batrachochytrium dendrobatidis*)

Allison Q. Byrne[1]*, Anthony W. Waddle[2,3], Veronica Saenz[4], Michel Ohmer[4,5], Jef R. Jaeger[3], Corinne L. Richards-Zawacki[4], Jamie Voyles[6], Erica Bree Rosenblum[1]

1 Department of Environmental Science, Policy, and Management, University of California Berkeley, Berkeley, California, United States of America, 2 One Health Research Group, Faculty of Veterinary and Agricultural Sciences, University of Melbourne, Victoria, Australia, 3 School of Life Sciences, University of Nevada, Las Vegas, Las Vegas, Nevada, United States of America, 4 Department of Biological Sciences, University of Pittsburgh, Pittsburgh, Pennsylvania, United States of America, 5 Department of Biology, University of Mississippi, Oxford, Mississippi, United States of America, 6 Department of Biology, University of Nevada Reno, Reno, Nevada, United States of America

* allie128@berkeley.edu

**Data Availability Statement:** All data used for this study and R code used for the analyses are available on GitHub: (https://github.com/allie128/Bd_NV_PA).

## Abstract

Host-pathogen specificity can arise from certain selective environments mediated by both the host and pathogen. Therefore, understanding the degree to which host species identity is correlated with pathogen genotype can help reveal historical host-pathogen dynamics. One animal disease of particular concern is chytridiomycosis, typically caused by the global panzootic lineage of the amphibian chytrid fungus (*Batrachochytrium dendrobatidis*, Bd), termed Bd-GPL. This pathogen lineage has caused devastating declines in amphibian communities around the world. However, the site of origin for the common ancestor of modern Bd-GPL and the fine-scale transmission dynamics of this lineage have remained a mystery. This is especially the case in North America where Bd-GPL is widespread, but disease outbreaks occur sporadically. Herein, we use Bd genetic data collected throughout the United States from amphibian skin swabs and cultured isolate samples to investigate Bd genetic patterns. We highlight two case studies in Pennsylvania and Nevada where Bd-GPL genotypes are strongly correlated with host species identity. Specifically, in some localities bullfrogs (*Rana catesbeiana*) are infected with Bd-GPL lineages that are distinct from those infecting other sympatric amphibian species. Overall, we reveal a previously unknown association of Bd genotype with host species and identify the eastern United States as a *Bd* diversity hotspot and potential site of origin for Bd-GPL.

## Introduction

When host-pathogen specificity exists within a system it can serve to reveal historical disease dynamics and has consequences for the evolution of virulence. Theoretical work predicts host-pathogen specificity will evolve, as a pathogen consistently exposed to a particular host would

**Funding:** The work in PA was supported by the US Department of Defense (SERDP: Project No. RC-2638) award to CRZ, JV, EBR. The work in NV was supported in part by the Bureau of Land Management under agreement L16AC00149 to JRJ at the University of Nevada Las Vegas.

**Competing interests:** The authors have declared that no competing interests exist.

be subject to stronger selection from that host and therefore evolve faster than a generalist pathogen [1]. However, the high frequency of multi-host pathogens in nature implies that interacting factors present in complex natural systems may tip the scales in favor of generalist pathogens [2]. When host-specificity is high, it is predicted that host-pathogen feedback will favor selection towards decreased pathogen virulence [3]. Similarly, low levels of specificity favor pathogens with high virulence and often lead to the competitive exclusion of other less fit pathogen strains [3]. Therefore, understanding the degree to which host-pathogen specificity exists in a system can shed light on the underlying evolutionary processes and disease dynamics at play.

Advances in genetic monitoring tools have allowed us to investigate fine scale disease dynamics in natural systems and uncover previously unknown host-pathogen genetic associations. The utility of DNA sequence data in understanding pathogen evolution and genetic diversity has been evidenced by genetic monitoring efforts during the Covid-19 pandemic (e.g., [4]. By applying technologies used in human pathogen systems to wildlife diseases we can investigate the epidemiology of diseases that are of conservation concern. Of particular interest is the amphibian chytrid fungus [*Batrachochytrium dendrobatidis* (Bd)] [5] because of its association with amphibian die offs and extinctions in many disparate areas of the world [6–10]. For Bd, there are at least five distinct genetic lineages, but the Global Panzootic Lineage (Bd-GPL) has been associated with almost all disease outbreaks around the world [11–13]. Recent DNA sequencing of Bd samples collected in North America has confirmed previous reports of two distinct sub-lineages within Bd-GPL, termed "Bd-GPL1" and "Bd-GPL2" [14–16]. The subclades Bd-GPL1 and Bd-GPL2 were originally delineated using differences in characteristic "loss of heterozygosity" (LOH) regions in the genome [14, 17]. Because Bd-GPL1 genomes lack LOH regions present in Bd-GPL2 genomes, it was hypothesized that Bd-GPL1 is the ancestral panzootic lineage [18]. While it has become increasingly clear that there is deep genetic structure within Bd-GPL, and that both Bd-GPL1 and Bd-GPL2 often co-occur, what remains unclear is how these patterns arose and are maintained across fine spatial scales.

A growing number of studies have shown that Bd, and specifically Bd-GPL, has patterns of genetic diversity that are linked to geography, but none have tested the hypothesis that there is a link between Bd genetics and host species [13, 15, 16]. In this study, we investigated the relationship between host species and Bd genotype in two distinct amphibian communities within the United States. One community was sampled in northwestern Pennsylvania near the Pymatuning Lab of Ecology (PA). Sampling in this temperate region took place at eight sites (including permanent and ephemeral ponds and one forested stream site). Seventeen different amphibian species were encountered in PA during this sampling period, including ten urodele and seven anuran species. The second amphibian community was sampled in southern Nevada (NV) within the eastern Mojave Desert in an area characterized by permanent water bodies separated by larger distances. Six different anuran species were encountered during field surveys in NV. While most amphibian species encountered in this study were found in only one of these regions, the American bullfrog [*Rana catesbeiana* (Shaw 1802), hereafter "bullfrogs"] occurred across both regions in either its native (PA) or introduced (NV) range. By comparing the relationship between Bd genotype and host species in these two localities we not only reveal more about the evolutionary history of Bd in the United States (US), but can also begin to understand the repeatability of these patterns across systems and the role certain amphibian species may have in shaping Bd genetic diversity.

## Materials and methods

### Sample collection in northwestern Pennsylvania and southern Nevada

We collected Bd samples in PA between 2017–2019 and in NV between 2016–2018. All work with amphibians was given written consent by ethics committees at either the University of Pittsburgh (IACUC protocol numbers 16027711 and 19024619) or the University of Nevada, Las Vegas (IACUC protocol number 870994). Additional ethics approval for work done in Pennsylvania was granted through the Department of Defense ACURO system (protocol number SERDP-RC-2638.03). Permission to sample in Pennsylvania was granted by the Pennsylvania Fish and Boat Commission. Permission to sample in Nevada was given by Bureau of Land Management, US Department of Interior (for Desert National Wildlife Range), Nevada Division of State Parks (for Spring Mountain Ranch State Park), and Nevada Department of Wildlife (for Lake Mead National Recreation Area).

In both areas, we captured frogs by hand using clean nitrile gloves. In PA we also used nets with recommended field hygiene protocols followed [19]. We kept captured amphibians individually in clean plastic bags during processing. To collect skin cell samples, we swabbed frogs in PA 5 times each on dorsal, ventral, right, and left sides, and feet with a sterile swab. In NV, we swabbed frogs 10 times on each ventral side and 5 times on each foot [20]. We initially stored all swabs in tubes on ice and promptly transferred them to a freezer at -20 ˚C until processing. We extracted DNA from swabs in PA using the Qiagen DNEasy Blood and Tissue kit (Qiagen, Valencia, USA), and from NV using PrepMan Ultra [21], following both manufacturer's protocols.

To generate pure cultures of Bd from the relict leopard frog (*Rana onca*) we used a non-lethal method in the field [22]. We excised a small piece of webbing tissue (1 mm$^2$) that was cleaned and embedded into TGhL agar plates containing a cocktail of antibiotics (kanamycin, ciprofloxacin, streptomycin, and penicillin) [23]. For the other species in NV from which cultures were derived, we brought the animals into the laboratory and euthanized by soaking in buffered MS222 followed by double pithing prior to excising 1 mm$^2$ pieces of skin tissue from the ventral abdomen, legs, and rear feet. Each tissue sample was then wiped through and embedded into a TGhL agar plate containing antibiotics. We established pure cultures by transferring Bd growth to H-broth liquid medium. We used a modified phenol-chloroform extraction to isolate DNA from the Bd cultures [24].

### DNA sequencing and preprocessing

We genotyped Bd from PA and NV using a custom genotyping assay [25]. For the PA samples, we initially sequenced 115 skin swabs and for the NV samples, we initially sequenced 28 pure isolates and 36 skin swabs. After downstream filtering for missing data (see below) our final dataset included 71 samples from PA, and 52 samples from NV. For some of our analyses we also included previously published Bd sequence data from samples collected from around the world [13].

We used the Fluidigm Access Array platform to perform microfluidic multiplex PCR on 191 regions of the Bd genome and one diagnostic locus for the closely related *Batrachochytrium salamandrivorans* [26]. Each target locus is 150–200 base pairs long, with the targets distributed across the Bd nuclear and mitochondrial genomes [25]. Extracted DNA from swab samples was cleaned and concentrated using an isopropanol precipitation. Isolate samples were diluted to a concentration of 3ng/μl using PCR-grade water. All samples were preamplified in two separate PCR reactions, each containing 96 primer pairs at a final concentration of 500nM. For each preamplification PCR reaction we used the FastStart High Fidelity PCR

System (Roche) at the following concentrations: 1x FastStart High Fidelity Reaction Buffer with $MgCl_2$, 4.5mM $MgCl_2$, 5% DMSO, 200µM PCR Grade Nucleotide Mix, 0.1 U/µl FastStart High Fidelity Enzyme Blend. We added 1µl of cleaned DNA to each preamplification reaction and used the following thermocycling profile: 95˚C for 10 min, 2 cycles of 95˚C for 15 seconds and 60˚C for 4 minutes, then 13 cycles of 95˚C for 15 seconds and 72˚C for 4 minutes. Preamplified products were treated with 4µl of 1:2 diluted ExoSAP-it (Affymetrix Inc.) and incubated for 30 min at 30˚C, then 15 min at 80˚C. Treated products were diluted 1:5 in PCR-grade water. The diluted products from each of the two preamplification reactions were combined in equal proportions and used for downstream amplification and sequencing. Each preamplified sample was then loaded onto a Juno™ LP 192.24 IFC (Fluidigm, Inc.) for library preparation. This flow cell performs microfluidic multiplex PCR to amplify all samples using 24 separate pools each containing 8 primer pairs. Barcoded samples were then pooled and sequenced on an Illumina MiSeq lane using the 300bp paired-end kits at the University of Idaho IBEST Genomics Resources Core.

### Sequence data analysis

We pre-processed all sequencing data as described in [25] and generated consensus sequences for all variants present for each sample at each locus using IUPAC ambiguity codes for multiple alleles. We filtered reads by selecting sequence variants that were present in at least 5 reads and represented at least 5% of the total number of reads for that sample/locus using dbcamplicons (https://github.com/msettles/dbcAmplicons).

We used a gene-tree to species-tree approach to construct phylogenies for the combined set of all sequenced samples and separately for the PA and NV sequence datasets. These trees allow us to explore the relationship of the Bd collected for this study to previously-published Bd samples representing all known Bd lineages [13, 25]. First, we trimmed our consensus sequence dataset to eliminate loci that had more than 50% missing data for either the NV or PA datasets, resulting in 174 loci. Next, for each dataset we individually aligned all loci using the MUSCLE package [27] in R (v.3.4.3), checked the alignments for errors in Geneious (v.10.2.3) [28], and used the RAxML plugin [29] in Geneious to search for the best scoring maximum likelihood tree for each locus using rapid bootstrapping (100 replicates). We then collapsed all branches in all trees with <10 bootstrap support and used Astral-III to generate a consensus tree [30]. Astral generates an unrooted species tree given a set of unrooted gene trees.

To further explore the genetic variation of Bd collected at our sites, we generated a dataset of single nucleotide polymorphisms (SNPs). First, we aligned all raw sample reads (PA and NV) and reads from 26 previously published Bd-GPL reference samples [13, 17] using bwa mem [31] to a reference fasta containing target sequences for each amplicon extracted from reference genome JEL423 (Broad Institute). We then used Freebayes (v.1.1.0) to call variants based on haplotypes [32] stipulating that variants only be called on the 174 amplicons that passed the 50% missing data cutoff described above. Variants were then filtered using vcftools (v.4.2) [33] to only include variants with a minor allele frequency > 0.01, quality > 30, less than 10% missing data, and minimum depth of 5. The final variant set included 129 binary SNPs with 3.8% missing data across all samples.

We then imported our SNPs into R using the vcfR package (v.1.12.0) [34] and converted the variants to a genlight object. To determine the optimal number of clusters in our data we used a Discriminant Analysis of Principal Components (DAPC) [35]. First, we ran the find. clusters function from the R package adegenet (v.2.1.1) with a maximum number of clusters = 10 [36]. This function implemented the clustering procedure used in DAPC by running successive K-means with clusters from K = 1 to K = 10 after transforming the data using a

principal component analysis (PCA). For each model BIC was computed as a measure of statistical goodness of fit. The best K was chosen as the value for K after which increasing K no longer leads to an appreciable improvement of fit (measured as substantial decrease in BIC) [35]. We then ran the DAPC analysis specifying the optimal number of clusters. To determine the optimal number of PCs to use for this analysis we ran the adegenet function "optim.a. score" which calculates the alpha score as $(P_t-P_r)$, where $P_t$ is probability of reassignment to the true cluster and $P_r$ is the probability of reassignment to randomly permuted clusters. We used a minimum of 2 PCs in the DAPC calculations. Finally, we conducted a PCA using the "glPCA" function in adegenet and plotted the PCA using the DAPC clusters to visually group samples together. Samples with posterior probability of inclusion in one cluster < 0.99 were labeled "unassigned." We plotted the first two PCs which together explain 68.2% of the variation in our data. We then computed a PCA separately for each locality to show within locality diversity across species and sites. To further characterize the genetic clusters identified using the DAPC analysis, we used the SNP data to calculate individual heterozygosity with vcftools (v.4.2) [33]. We then used a Wilcox test to compare mean individual heterozygosity between genetic clusters using the "geom_signif" function in the ggsignif (v.0.6.3) R package [37].

To evaluate the relative contribution of sampling site and host species we ran an analysis of molecular variance (AMOVA) [38] using our SNP data. This analysis was conducted separately for the PA and NV datasets and determined the relative influence of within individual variation, host species identity, and individual sampling locality on Bd genetic structure. For the AMOVA, we included within individual variation as a null hypothesis, as you would expect a panmictic population to have a large proportion of genetic variation arising within individuals rather than between localities or host species. We ran this analysis using the poppr package (v.2.9.0) in R [39] and only used SNPs with <5% missing data. This resulted in 40 high quality SNPS for PA and 126 high quality SNPs for NV. We then tested for statistical significance using the "randtest" function in the package ade4 (v.1.7–16), randomly permuting our data 999 times. To further characterize the relationship of geography and Bd genetic diversity, we calculated pairwise genetic distance between samples using the "bitwise.dist" function from the poppr R package [39], and pairwise geographic distance in kilometers in R. We then used a mantel test, as implemented in the vegan R package (v.2.5–7) [40], to test for correlation between the genetic and geographic distance matrices.

To evaluate the frequency of mixed infections on individuals, we searched our set of amplicon sequences for those that had diagnostic alleles that discriminate between Bd-GPL1 and Bd-GPL2 lineages. First, we searched the amplicon sequences from the set of 26 pure culture, reference samples. We identified amplicons that had at least 1 SNP that distinguished GPL1 samples from GPL2 (as defined by both the DAPC and phylogenetic analyses described above). The GPL1 Bd isolates in this reference set were CJB4, JEL238, JEL267, JEL359, MexMkt, MLA1, SRS812, TST75. The GPL-2 Bd isolates were Campana, CLB5-2, CJB7, EV001, JAM81, JEL271, JEL275, JEL310, JEL427, JEL429, JEL433, JEL627, Lb_Aber, NBRC106979, Pc_CN_JLV, and RioMaria. These samples were all sequenced from pure cultures grown in the lab, and previously-published whole genome data indicates these are pure Bd cultures representing a single lineage [17, 41]. From these samples we identified 24 loci that occur on five separate Bd chromosomes that have specific Bd-GPL1 or Bd-GPL2 alleles. For the diagnostic alleles on chromosome 1, Bd-GPL2 isolates can be either homozygous for the Bd-GPL2 allele or heterozygotes, while Bd-GPL1 isolates are always homozygous for the Bd-GPL1 allele (see S1 Fig in S1 File). We then manually scored each sample from PA and NV at each as having either the Bd-GPL1, Bd-GPL2, or both alleles present (heterozygote).

All data used for this study and R code used for the analyses are available on github: (https://github.com/allie128/Bd_NV_PA).

## Results

We found a repeated pattern of Bd genetic structure that is most closely associated with amphibian host species in both PA and NV. The DAPC analysis of the combined PA and NV Bd genetic data show that two clusters (K = 2) was the most likely grouping for our data (as inferred from ΔBIC, S2B Fig in S1 File). In the combined PCA, we see a clear division of Bd genotypes into two clusters, with the presence of intermediate "unassigned" samples (Fig 1C). By evaluating each site separately, we see that PA harbors much more genetic variability within samples (accounting for all "unassigned" Bd genotypes, Fig 1). We confirmed that "unassigned" samples were not simply the product of missing data by showing that each genotype group (Bd-GPL1, Bd-GPL2, and unassigned) had the same distribution of missing data (S2A, S2C Fig in S1 File).

Further phylogenetic analyses incorporating previously published Bd isolates shows that that all Bd samples sequenced for this study belong to the Bd-GPL (Figs 2 and 3 and S3 Fig in S1 File) and that the two clusters identified in our dataset correspond to the previously published Bd-GPL subclades Bd-GPL1 and Bd-GPL2 [14, 18]. The difference between Bd-GPL1 and Bd-GPL2 samples is shown clearly in the combined PCA (Fig 1C), where PC1 (which explains 58.4% of the variation in our genetic data) separates samples into each of these subclades. In NV, 20 samples were assigned to Bd-GPL1 and 32 samples were assigned to Bd-

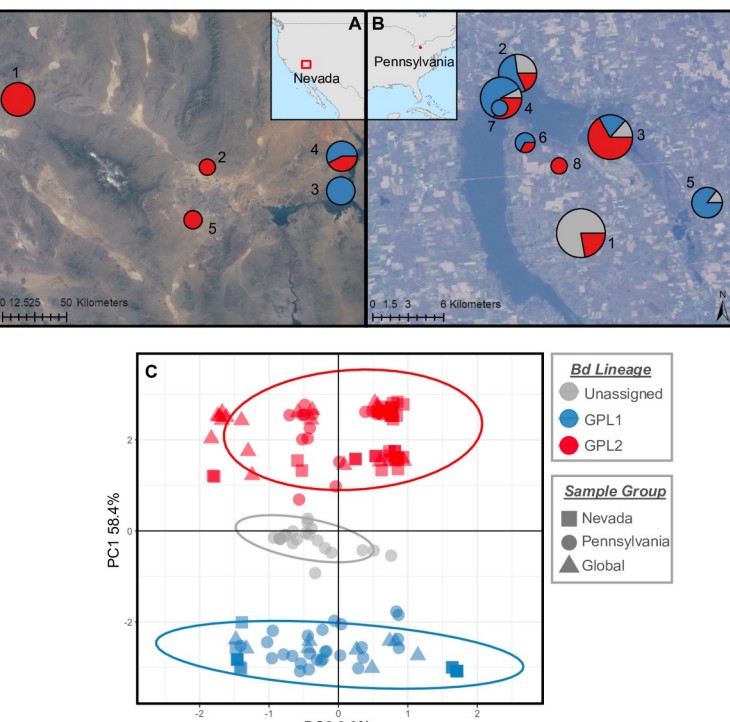

**Fig 1.** Map of sampling localities for Bd in southern Nevada (A) and northwestern Pennsylvania (B). Individual sampling localities are numbered and represented with a pie chart that shows the proportion of samples belonging to each Bd-GPL sub-lineage as determined using the DAPC clustering analysis. Unassigned samples (in grey) are those with a posterior probability of assignment to either group < 0.99. Pie charts are scaled in size according to the number of samples collected at that site. (C) PCA calculated from 129 SNPS using the combined dataset of southern Nevada (N = 52, squares), northwestern Pennsylvania (N = 71, circles), and global reference samples (N = 26, triangles). Symbols and ellipses colored as in maps above. Satellite imagery courtesy of the Earth Science and Remote Sensing Unit, NASA Johnson Space Center (http://eol.jsc.nasa.gov).

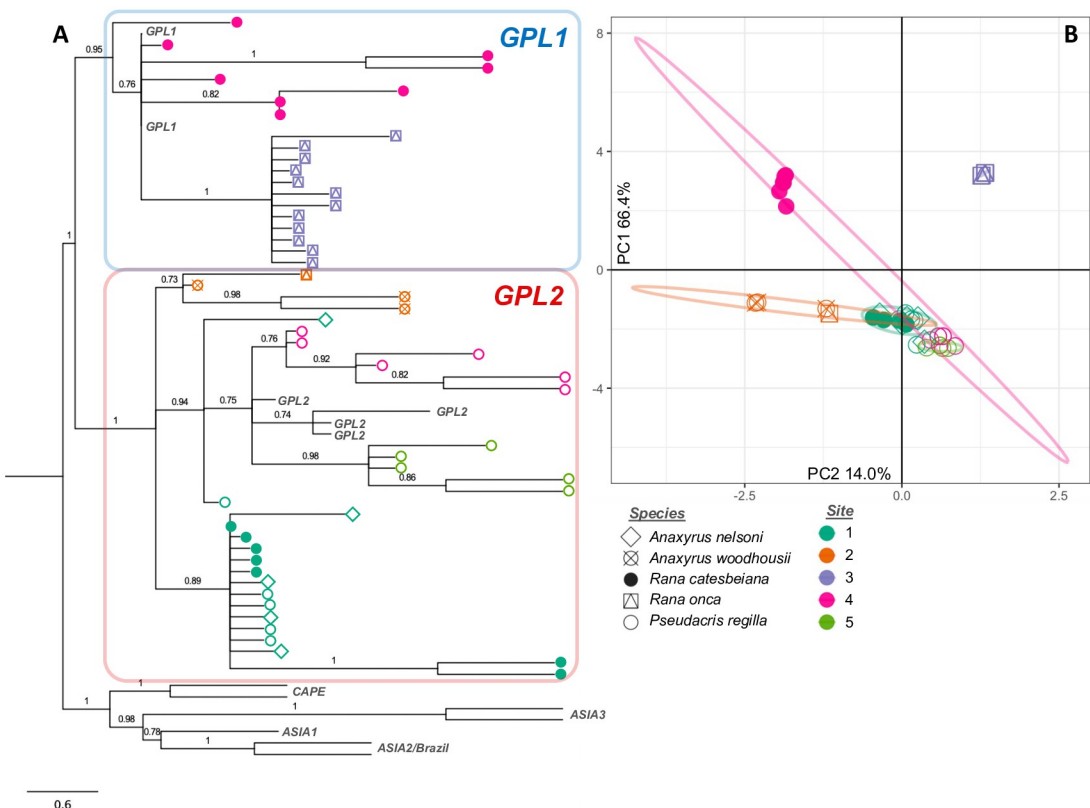

**Fig 2. (A) Phylogenetic tree and (B) PCA for 52 Bd samples collected in southern Nevada.** Colors represent the locality where the Bd was sampled (numbered as in Fig 1) and symbols represent host species. Tree is calculated using Astral-III and includes and additional 13 reference samples representing all known clades of Bd. Nodes are labeled with posterior probability and those with posterior < 0.7 have been collapsed.

GPL2 (Fig 2). In PA, 27 samples were assigned to Bd-GPL1, 23 samples were assigned to GPL-2, and 21 samples we unassigned (Fig 3).

The factors that may explain Bd genetic structure at each site were determined using an AMOVA. We found that most of the variation in Bd genotypes (64.3% in PA, 58.4% in NV) can be explained by variation between host species, rather than within an individual (34.7% in PA, 8.8% in NV) or between sampling sites (1.0% in PA, 32.8% in NV). Comparisons of variation within individuals and variation between host species were statistically significant using permutation tests to randomly permute samples between groups (999 times, p < 0.001 for each test, S3 Fig in S1 File). We also found a weak, yet statistically significant association between geographic and genetic distance using a mantel test in both NV (r = 0.39, p = 0.001) and PA (r = 0.13, p = 0.002) (S4 Fig in S1 File).

We can evaluate each individual sample for signatures of coinfections or the presence of a hybrid lineage (i.e., the presence of both Bd-GPL1 and Bd-GPL2) by looking at Bd-GPL1/2 diagnostic alleles and comparing patterns of heterozygosity between genotypes. The allelic patterns show that the "unassigned" samples in PA appear to either be from a mixed infection or infection with a hybrid lineage—that is, they show a high frequency of heterozygosity (S6 Fig in S1 File) and/or the presence of alleles from different Bd-GPL subclades at genomically proximate loci (Fig 4). In PA, one site (PA Site 1) accounts for most of the unassigned samples (67%, 14/21); additionally, 71% (15/21) of unassigned Bd samples were collected from green

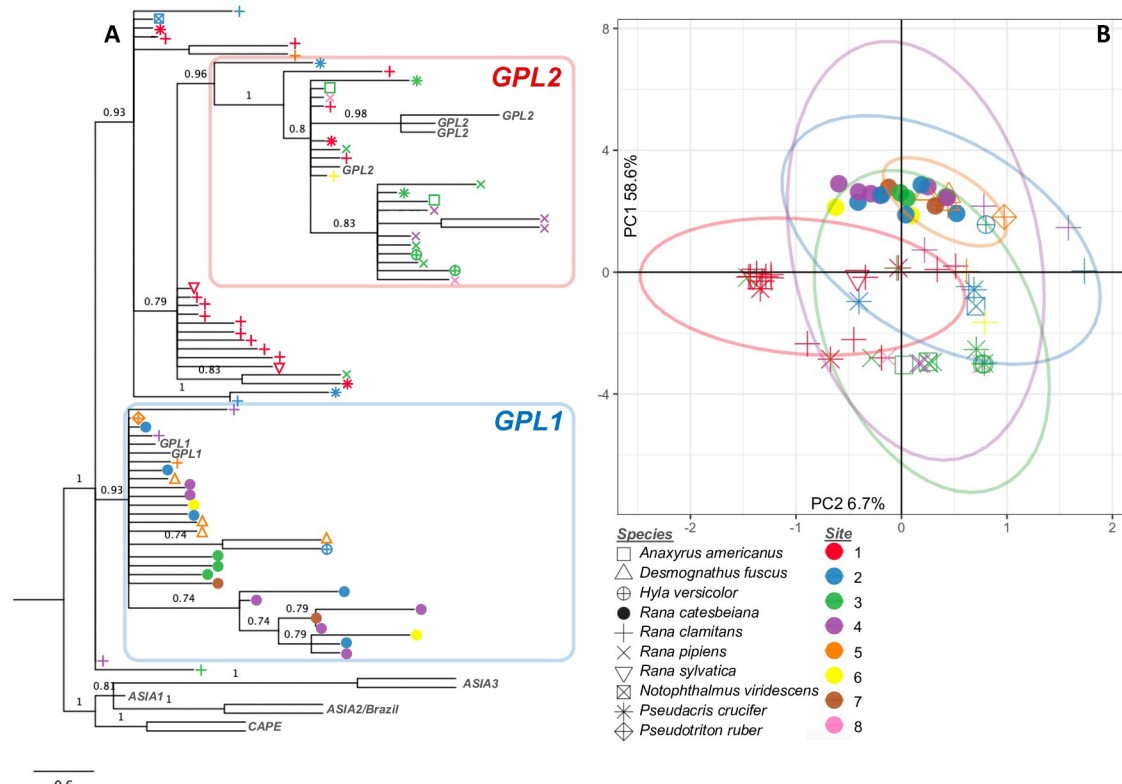

**Fig 3. (A) Phylogenetic tree and (B) PCA for 71 Bd samples collected in northwestern Pennsylvania.** Colors represent the locality where the Bd was sampled (numbered as in Fig 1) and symbols represent host species. Tree is calculated using Astral-III and includes and additional 13 reference samples representing all known clades of Bd. Nodes are labeled with posterior probability and those with posterior < 0.7 have been collapsed.

frogs (*Rana clamitans*). Furthermore, all 18 samples collected from bullfrogs in PA were assigned to GPL-1, despite being collected at five different localities, each with multiple Bd-GPL genotypes present (Figs 3 and 4).

In NV we see very little Bd-GPL1/Bd-GPL2 mixing within samples and a strong association of Bd-GPL1 with bullfrogs at one site (NV Site 4), and *R. onca* at another (NV Site 3). While *R. onca* was the only species caught at NV Site 3, bullfrogs were caught alongside Pacific tree frogs (*Pseudacris regilla*) at NV Site 4 and these two species were host to distinct Bd-GPL genotypes with virtually no allele mixing (Fig 4). At another site (NV Site 1), however, bullfrogs were host to Bd-GPL2 as were all other amphibian hosts at that locality. Additionally, Bd-GPL2 found at NV Site 1 shows a pattern of high heterozygosity at diagnostic alleles found on Bd Chromosome 1, a pattern similar to the reference Bd-GPL2 isolates JEL627, JEL271, Lb_A-ber, CJB7, and JAM81 (S6 Fig in S1 File).

## Discussion

Herein, we show a clear, repeated pattern of Bd genetic diversity associated with host species at single geographic sites. While previous studies have shown some broad geographic patterning in Bd genetic structure [15, 16], this is the first study to link fine-scale Bd genetic variation to host species identity. Furthermore, this pattern is repeated across individual sites in PA, and in two separate regions of the US. Remarkably, the same association of Bd-GPL1 with bullfrog

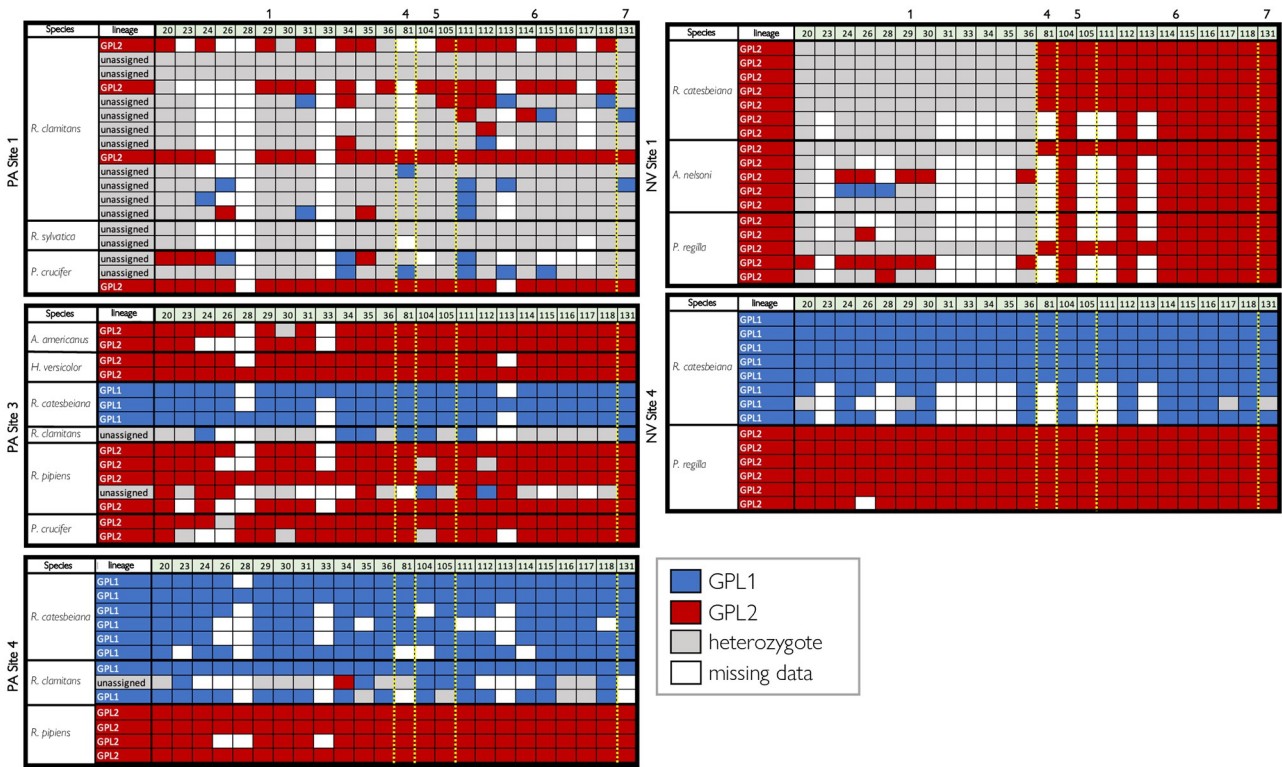

**Fig 4. Alleles for PA Site 1 (left top), PA Site 3 (left middle), PA Site 4 (left bottom), NV Site 1 (right top), and NV Site 4 (right bottom).** Each row represents an individual sample, and each numbered column represents a nuclear locus that has an informative GPL1/GPL2 allele (N = 24). Loci are organized according to position along the Bd genome, with the yellow dashed indicating a change to a new chromosome and the chromosome number labeled along the top. Samples are grouped by host species ("species" column) and labeled with the DAPC assigned lineage ("lineage" column). A red box indicates a Bd-GPL2 allele for that sample at that locus, a blue box represented a Bd-GPL1 allele, a gray box indicates both alleles are present, and a white box indicates missing data. Representative sequences for each allele are included in the supplemental materials.

hosts was seen in both PA and NV. Below, we explore how this pattern may have arisen in each region, discuss what this may mean for the origin of Bd-GPL, and explore the potential importance of genetically diverse Bd infections.

## How did host-pathogen specificity arise in each region?

Processes occurring at different timescales could produce the patterns of Bd genotype and host species association observed in each region. One possibility is that long-term coevolutionary relationships between Bd and its host species triggered *in situ* evolution and subsequent pathogen divergence over time in the ancestral Bd-GPL population. This process could have been mediated by many factors including host immune systems, skin microbial communities, and the environment. Furthermore, mutational changes in the pathogen that have consequences for infectiousness across different hosts could contribute to evolution towards host specificity [42]. Bd can suppress host immune responses and some susceptible species show an over-activation of the immune system that may worsen disease outcomes [43]. At the same time, immunization against Bd has been successful in some species, indicating the relationship between adaptive immunity and Bd varies between hosts [44]. Amphibian species identity is often the strongest predictor of skin microbial community composition [45, 46] and these microbes can be associated with Bd disease outcomes [7, 47, 48], providing another possible

mechanism for facilitating the evolution of host-pathogen specificity. Finally, Bd isolates in the lab can show rapid evolutionary change over short time periods [49, 50]. Therefore, differences between Bd genotypes in the way they interact with amphibian skin microbes or immune systems, in combination with Bd's evolutionary lability, could allow for the formation of the host-pathogen genetic correlations that we detected.

In PA, we observed a ubiquitous pattern of association between Bd-GPL1 and bullfrogs in all individual sites. Moreover, not a single bullfrog sample (N = 18) in the PA region was infected with Bd-GPL2 or included in the "unassigned" group. The finding that host species was more important than geography in explaining Bd genetic patterns in PA has not been documented in other regions and may indicate that Bd-GPL has been present for a long time in this region and has coevolved with native amphibians, including bullfrogs. We can see additional evidence for this hypothesis at individual sites; at PA Site 4 bullfrogs and the closely related green frog (*Rana clamitans*) were both preferentially infected with Bd-GPL1, while the more distantly related northern leopard frog (*Rana pipiens*) was infected with Bd-GPL2 [51]. This pattern follows other studies showing that specialized pathogens are more likely to infect closely related host species [52]. Overall, when comparing patterns of Bd genotypes and host species in PA to those documented in other areas of the world, we see preliminary evidence for a coevolutionary mechanism shaping Bd-GPL genetic diversity.

An alternative possibility that could produce patterns of host species-Bd genotype correlations is a recent introduction of a novel Bd-GPL lineage into an amphibian community. Existing differences in the ability of the novel Bd-GPL lineage to infect different hosts could maintain the stratification of Bd genotypes along species boundaries. In NV the association between Bd genotypes and host species is not as consistent as in PA, indicating the coexistence of different Bd genotypes in the NV region may be a more recent phenomenon. For example, at NV Site 4 Bd genotypes were strictly partitioned by species, with Bd-GPL1 only present on bullfrogs; however, at NV Site 1 all individuals sampled across species were infected with the same Bd-GPL2 genotype, including bullfrogs. Bullfrogs are not native to Nevada and have been present since at least 1933, likely introduced during fish stocking or from the food trade [53]. Therefore, in this region, bullfrog colonization may have provided for a more recent introduction of Bd-GPL1. At NV Site 1 Bd-GPL2 may have outcompeted other Bd lineages that could have been present when bullfrogs were first introduced. Supporting this hypothesis, one study found that bullfrogs farmed in Idaho and used for an infection experiment initially harbored cryptic, low intensity infections of Bd-GPL1 that were eventually outcompeted by an inoculation with a high dose of a Bd-GPL2 isolate [54]. Importantly, the NV sites are home to a much less diverse amphibian community than those in PA, perhaps providing fewer ecological opportunities for Bd and therefore a less diverse Bd community overall. What host and environmental factors allow for the replacement of one Bd lineage by another and/or Bd lineage coexistence is not altogether clear but is a key question going forward to better understand the role of potential disease vectors such as the bullfrog.

Future studies addressing these two hypotheses could use a variety of approaches. For example, future experiments could use non-lethal mucosome assays, which are a reliable predictor of Bd infection in the field [55], to compare mucosal function of different amphibian host species against a genetically diverse array of Bd-GPL isolates. This research could investigate interactions between bullfrog mucosal skin communities and different Bd lineages, in addition to other potentially important host species in the US. For example, we found that green frogs (*R. clamitans*) had a high frequency of mixed infections and chorus frogs (*P. regilla*) were exclusively infected with Bd-GPL2 in Nevada. Intriguingly, *P. regilla* have previously been proposed as important Bd reservoirs in the Western USA [56, 57], further motivating future research on this species. Additionally, comparing host and pathogen genomics

using cophylogenetic methods could provide important insights into potential cospeciation and/or coevolution that has occurred [58]. A better understanding of the role of host species in shaping Bd community composition can provide insights into the evolutionary history of Bd and inform future management for this globalized pathogen.

## What does our data tell us about the site of origin for Bd-GPL?

The implied coexistence of Bd and amphibians at evolutionary time scales in PA could indicate that the source population of the most recent common ancestor of modern Bd-GPL lineages existed in this part of the world. This possibility has been previously proposed, following reports of higher diversity in Bd samples collected from bullfrog populations [59, 60], and evidence that bullfrogs may have facilitated Bd spread from east to west in the US [61]. Adding to evidence that North America may be site of origin for Bd-GPL, a recent study found as much genetic variation in Bd-GPL samples collected from the Sierra Nevada as was present in a large global sample of Bd-GPL genotypes [15]. In other regions, such as Panama and Mexico, genetic diversity of Bd-GPL is either spatially structured across the region (Mexico) or genetically homogenous (Panama), the latter indicating a more recent introduction and spread [15, 16]. Studies using swabs collected from museum specimens have found Bd in Illinois as early as 1888 [62] and in bullfrogs collected in California in 1928 [63]. While there is a growing body of evidence suggesting that North America, and perhaps the eastern US (within the native range of the bullfrog) may have been the site of origin for Bd-GPL, more studies are needed to investigate this hypothesis. It will be particularly important to include more Bd genetic data from Europe, where there is also high genetic diversity within Bd-GPL [11].

## What can the "unassigned" samples tell us about Bd dynamics?

One key question raised from our data concerns the Bd genotypes that were "unassigned" and lie in between Bd-GPL1 and Bd-GPL2 genotypes in our PCA. These samples, all of which were from swabs collected in PA, show high levels of heterozygosity across the entire panel of diagnostic alleles, and significantly higher heterozygosity overall. There are two possible explanations for the occurrence of a highly heterozygous, intermediate Bd genotype. The first is that these individuals were coinfected with both Bd-GPL1 and Bd-GPL2. The second is that they were infected with a hybrid lineage resulting from recombination between Bd-GPL1 and Bd-GPL2. Indeed, if we look at the reference Bd isolates in S3 Fig in S1 File we see that some Bd-GPL2 pure isolates are heterozygous at certain alleles, indicating that high heterozygosity could result from a single infection. Herein, we are unable to confidently say whether these ambiguous samples represent a mixed infection or an infection with a hybrid lineage.

Our data show not only the coexistence of multiple Bd-GPL lineages at a single site, but also highlights areas where Bd lineage mixing on individuals is rare (in NV, and PA Sites 3 and 4), or common (PA Site 1). The most common frog species caught at PA Site 1 was the green frog (*R. clamitans*), in fact this species accounts for 71% of all unassigned samples in this study. At this point it is unclear whether there is something unique about PA Site 1, or *R. clamitans*, that may allow for such consistent lineage mixing or the potential presence of hybrid lineages. To our knowledge no chytridiomycosis outbreaks have been reported in *R. clamitans*, and individuals from the same site in PA showed no increase in stress hormones when experimentally infected with Bd [64]. Probably, this species is a tolerant carrier of Bd, like the closely related bullfrog. This host-pathogen relationship is worth further exploration as the potential for Bd hybridization could lead to novel pathogen phenotypes.

The frequency and circumstances under which recombination occurs in Bd is still unclear. There is, however, plenty of genetic evidence that recombination has occurred between

divergent Bd lineages [11, 14, 65]. A possible mechanism for genomic recombination in Bd is parasexual reproduction—where diploid spores fuse to form tetraploid progeny, after which chromosomes are lost as the organism returns back to a diploid state [66]. This process has been documented in the human fungal pathogen *Candida albicans*, although the use of this reproductive mode seems to be restricted in nature [67]. Parasexual reproduction often results in aneuploidy, which is a common phenomenon in Bd [17, 68], however known hybrids between Bd lineages did not have more than two alleles at any sequenced loci in previous studies [14, 65]. Furthermore, elevated chromosomal copy number has been linked to increased virulence in Bd [68], and hybrids of Bd have been more deadly than parent lineages when paired with certain host species [69]. Indeed, it was originally proposed that Bd-GPL could have originated from recombination between closely related Bd lineages [12], but this hypothesis has been a topic of much debate [70]. Most researchers on the subject, however, would agree that precautions should be taken to reduce unintended spread of Bd to limit the possibility of recombination that could produce more deadly lineages.

## Conclusions

Herein we document an association between Bd-GPL genotypes and host species in two regions of the US. We believe the patterns seen in PA may be the result of a long-term coevolutionary relationship between Bd-GPL1 and bullfrogs (and perhaps other closely related species such as the green frog). We observed that the association of Bd-GPL1 with bullfrogs is consistent in NV where bullfrogs have been more recently introduced, and that the patterns in this region are consistent with more recent introductions of novel Bd lineages. We also found evidence of either mixed Bd infections or a potential hybrid lineage that was common at a single site in PA and on green frogs across all PA sites. Given the high genetic diversity and evidence of evolutionary relationships between Bd-GPL lineages and particular host species, we posit that North America may be the site of origin for modern Bd-GPL lineages. Overall, this study presents evidence that Bd-GPL has a long history of coexistence with amphibians in the US, reveals more about the relationship of the bullfrog to Bd, and points to a possible cradle of Bd-GPL genetic diversity in the eastern US.

## Supporting information

**S1 File.**
(PDF)

## Acknowledgments

We thank Frank van Breukelen, Mark Slaughter, and Jonathan Smith for their assistance with funding acquisition and management for the Nevada study. We thank the following people for their assistance with field and laboratory work in Nevada: Rebeca Rivera, Daniel Villanueva, Yesenia Vasquez, David Miller, Joshua Levy, Kevin Guadalupe, Marlai Sai, Ghazal Rezaei, Alexandra Zmuda, Jessica Hill, Shaylene Scarlett, and Alexa Krauss. We thank David Spicer for access to Boiling Pot Ranch in Nevada. We thank the following people for assistance with field and laboratory work in Pennsylvania: Laura Brannelly, Karie Altman, Talisin Hammond, Jeffery Bednark, Paradyse Blackwood, Robert Campion, Lauren Chronister, Jordan Coscia, Nina Dunnell, Conor Harrington, Ally Hartman, Marcus Hough, Jennifer Kassimer, Stephanie Kubik, Sadie Parker, Natalie Popielski, Phoebe Reuben, Ayla Ross, Samantha Shablin, Samantha Skerlec, Trina Wantman, Lydia Zimmerman, Jakub Zegar, and Aimee Danley.

## Author Contributions

**Conceptualization:** Allison Q. Byrne, Veronica Saenz, Jef R. Jaeger, Corinne L. Richards-Zawacki, Erica Bree Rosenblum.

**Data curation:** Allison Q. Byrne, Anthony W. Waddle, Veronica Saenz, Michel Ohmer, Jef R. Jaeger.

**Formal analysis:** Allison Q. Byrne.

**Funding acquisition:** Jef R. Jaeger, Corinne L. Richards-Zawacki, Jamie Voyles, Erica Bree Rosenblum.

**Investigation:** Allison Q. Byrne, Anthony W. Waddle, Veronica Saenz, Michel Ohmer, Jef R. Jaeger, Corinne L. Richards-Zawacki, Erica Bree Rosenblum.

**Methodology:** Allison Q. Byrne, Anthony W. Waddle, Veronica Saenz, Michel Ohmer, Jef R. Jaeger, Corinne L. Richards-Zawacki, Erica Bree Rosenblum.

**Project administration:** Allison Q. Byrne, Michel Ohmer, Jef R. Jaeger, Corinne L. Richards-Zawacki, Erica Bree Rosenblum.

**Resources:** Erica Bree Rosenblum.

**Software:** Allison Q. Byrne.

**Supervision:** Allison Q. Byrne, Corinne L. Richards-Zawacki, Jamie Voyles, Erica Bree Rosenblum.

**Validation:** Allison Q. Byrne.

**Visualization:** Allison Q. Byrne.

**Writing – original draft:** Allison Q. Byrne, Erica Bree Rosenblum.

**Writing – review & editing:** Allison Q. Byrne, Anthony W. Waddle, Veronica Saenz, Michel Ohmer, Jef R. Jaeger, Corinne L. Richards-Zawacki, Jamie Voyles, Erica Bree Rosenblum.

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
