## [Decision Letter · Decision Letter 0]

5 Jan 2022

PONE-D-21-37079Host Species is Linked to Pathogen Genotype for the Amphibian Chytrid Fungus (Batrachochytrium dendrobatidis)

PLOS ONE

Dear Dr. Byrne and Co-authors,

Thank you for submitting your manuscript to PLOS ONE. After careful consideration, we feel that it has merit but does not fully meet PLOS ONE’s publication criteria as it currently stands. Therefore, we invite you to submit a revised version of the manuscript that addresses the points raised during the review process.

I apologize for the delay in a decision on you paper.  For reasons that are unclear to me, I found it difficult to secure good reviewers.  I contacted many potential reviewers, and many of them declined to review.  Your manuscript has now been reviewed by two knowledgeable reviewers, and both are generally positive.  Thus, I recommend minor revisions to address any reviewer concerns.

Please carefully read their comments and respond with a revised manuscript that addresses as many of the comments as possible. Provide your responses in a point-by-point Response to Reviewers.

We look forward to receiving your revised manuscript.

Kind regards,

Louise A. Rollins-Smith

Academic Editor

PLOS ONE

Journal Requirements:

3. To comply with PLOS ONE submissions requirements, please provide methods of sacrifice in the Methods section of your manuscript.

5. We note that Figure 1 in your submission contain map/satellite images which may be copyrighted. All PLOS content is published under the Creative Commons Attribution License (CC BY 4.0), which means that the manuscript, images, and Supporting Information files will be freely available online, and any third party is permitted to access, download, copy, distribute, and use these materials in any way, even commercially, with proper attribution. For these reasons, we cannot publish previously copyrighted maps or satellite images created using proprietary data, such as Google software (Google Maps, Street View, and Earth). For more information, see our copyright guidelines: http://journals.plos.org/plosone/s/licenses-and-copyright.

a) You may seek permission from the original copyright holder of Figure 1 to publish the content specifically under the CC BY 4.0 license.  

Reviewers' comments:

Reviewer's Responses to Questions

**Comments to the Author**

1. Is the manuscript technically sound, and do the data support the conclusions?

Reviewer #1: Yes

Reviewer #2: Partly

2. Has the statistical analysis been performed appropriately and rigorously? 

Reviewer #1: Yes

Reviewer #2: No

3. Have the authors made all data underlying the findings in their manuscript fully available?

Reviewer #1: Yes

Reviewer #2: Yes

4. Is the manuscript presented in an intelligible fashion and written in standard English?

Reviewer #1: Yes

Reviewer #2: Yes

5. Review Comments to the Author

Reviewer #1: Byrne et al. examine the population genetic structure of a fungal pathogen of amphibians from two regions of North America using a SNP genotyping method (Fluidigm). They sampled multiple species from a handful of sites in both Pennsylvania and Nevada. All of the samples are of the global panzootic lineage (GPL), and the authors are able to divide the genotypes into GPL-1, GPL-2 or “unassigned”. Unassigned genotypes are either a coinfection by both a GPL-1 & GPL-2 or a hybrid genotype. A major finding is that genotypes associated with bullfrogs are more often GPL-1, and genotypes from green frogs are more often “unassigned”. This leads to the conclusion that there is some level of specificity between host and pathogen genotypes and possibly coevolution.

Overall this paper is technically sound and presents a lot of new data. It also is a nice investigation of population structure and may change the way people think about the importance and role of Bd genotype in local disease dynamics. Most of my comments are about clarifications and interpretations.

Suggestions:

1. I think it is worth explaining what the basis for the GPL-1 vs. GPL-2 split is in the introduction and possibly discussion sections. GPL is really a clonal lineage that has fixed heterozygosity of some markers. GPL-1 is a version that has not had some consistent loss of heterozygosity (LOH) events that define GPL-2. A lot of these differences can be obscured by independent LOH events, and there may be intermediate genotypes that have only had some but not all of the LOH events. This paper is interesting in that it clearly picks up the two major groups and the intermediate. Ultimately it will be really fascinating if the unassigned genotypes in this study are sexually produced hybrids of these two sublineages. One thing worth noting is that European genotypes have largely been ignored in many pop gen studies, and I think it is important to consider that Europe may actually be the original location from which GPL dispersed across the world. See the phylogeny in Rosenblum 2013 which shows European strains at the base of the GPL clade.

2. There are a few places where terms like “potential ancestral population of GPL” are used, such as line 36 of the abstract. O’Hanlon and Byrne have shown that Asia has the highest amount of lineage diversity and is the most likely source of the GPL for that reason. At this point there is no evidence of any genotypes other than GPL in North America, but it is unclear when GPL might have colonized the region. Perhaps the authors are thinking about the source population of the MRCA of the known GPL strains in the world. As stated in point 1, I think we don’t know but I also think that you have to carefully consider European samples and collect more GPL genotypes from Asia before concluding that the GPL MRCA was North American.

3. I mentioned fixed heterozygosity in point 1. Do the fluidigm markers allow heterozygosity estimation? If so, it would be interesting to see average observed heterozygosity across GPL-1, GPL-2 and unassigned. If the markers allow phasing and haplotype estimation shouldn’t it be possible that some of the markers in the unassigned would show more than 2 haplotypes under coinfection and never more than 2 for the hybrid hypothesis? I realize the authors do mention parasexual reproduction in the Discussion, and that could lead to more than 2 alleles at an individual locus. I think it’s fine to mention parasexual reproduction, but it is perhaps worth mentioning that the hybrid genotypes reported in Schloegel and Jenkinson have never had more than 2 alleles per locus.

4. Line 47, some wording seems weird to me. Increased virulence will never be advantageous for a host. I think it may be more appropriate to say that strong host-specificity should favor intermediate virulence for that host.

5. Line 69, delete “remains untested”

6. Line 86, what kind of gloves?

7. Line 141, what is “(N=13; 13,25)”

8. Line 198-199, I think this should be clarified because as written it’s not clear why GPL-2 isolates would be heterozygous.

9. Line 247-248. Please explain what the test for within individual variation means. Variation is significantly structured by individuals?

10. Figure S2, please explain what the X axis represents.

11. The associations in the PA population are interesting. These data really suggest future experimental studies. It might be worth expressing how you might test what is happening here with bullfrog GPL-1 association experimentally.

12. Line 382, suggest “possible” instead of “likely”. In most senses parasexual and sexual reproduction yield a pretty similar outcome. It could be hard to distinguish the two without observing intermediates or complete genome sequences.

13. Line 386-387. This sentence was hard for me to follow. The use of parasexuality in nature is to present complex gene sets?

Reviewer #2: This study describes the genetic variation of a batrachochytrid fungus lineage (Bd-GPL) across several amphibian hosts in two regions of North America to explore pathogen diversification concerning host species. While the manuscript tackles an important topic, is well written, and provides detailed methods, in my opinion, the results are a little bit misleading, not providing enough support for the authors’ conclusions.

My main concern is that the sub-lineage category is ignored in the AMOVA analyses. Sub-lineage information might affect and confuse the relative contribution of host species to the genetic variance. In the data reduction analysis, the authors find that the sub-lineage explains the highest variation among samples (58.4%, Figure 1C), but this information is ignored in the subsequent analyses. Also, the PCA analysis clusters together samples from the two studied locations (Nevada and Pennsylvania), which, in turn, could change the interpretation of the results. A phylogenetic analysis integrating both regions could also improve result interpretation.

Likewise, when describing the phylogenetic clades would be important to better characterize them. For instance, genetic diversity statistics (e.g., nucleotide diversity) could be computed between each sub-lineage to compare regions. Moreover, the coevolutionary hypothesis should be supported with further analysis of co-phylogeny congruence between pathogen and host.

Another of my concerns is that Pennsylvania sampling sites could be dependent due to the geographic proximity. I recommend exploring the correlation between genetic and geographic distances (Mantel test).

Finally, I think that one overlooked difference between Pennsylvania and Nevada sites that affects the pathogen diversification is the amphibian community structure. It would be worthwhile to characterize and integrate amphibian community metrics (e.g., richness per site) in this study.

6. PLOS authors have the option to publish the peer review history of their article (what does this mean?). If published, this will include your full peer review and any attached files.

Reviewer #1: **Yes: **Tim James

Reviewer #2: No

---

## [Author Response · Author response to Decision Letter 0]

28 Feb 2022

Note that the line numbers listed below refer to the track changes version of the manuscript. 

Editor Comments:

The manuscript has been carefully reformatted to match the PLOS ONE style guide.

Corrected as suggested. We now include specific information about the permits granted for this work (Lines 131-135).

3. To comply with PLOS ONE submissions requirements, please provide methods of sacrifice in the Methods section of your manuscript.

We added this additional detail to the methods section (Lines 152-153)

Corrected as suggested. We now include additional details about the ethics approvals received from each institution in the methods section (Lines 126-131)

5. We note that Figure 1 in your submission contain map/satellite images which may be copyrighted. All PLOS content is published under the Creative Commons Attribution License (CC BY 4.0), which means that the manuscript, images, and Supporting Information files will be freely available online, and any third party is permitted to access, download, copy, distribute, and use these materials in any way, even commercially, with proper attribution. For these reasons, we cannot publish previously copyrighted maps or satellite images created using proprietary data, such as Google software (Google Maps, Street View, and Earth). For more information, see our copyright guidelines: http://journals.plos.org/plosone/s/licenses-and-copyright.

a) You may seek permission from the original copyright holder of Figure 1 to publish the content specifically under the CC BY 4.0 license. 

We changed the imagery used in Figure 1, sourcing the satellite images from the Gateway to Astronaut Photography of Earth. We thank the editor for pointing us to this public domain resource. We added an acknowledgement for the photos in the Figure 1 legend (Lines 307-308). 

We reviewed the references and edited them to match PLOS ONE’s guidelines (Lines 585-1343)

Reviewer Comments

Reviewer #1: 

Byrne et al. examine the population genetic structure of a fungal pathogen of amphibians from two regions of North America using a SNP genotyping method (Fluidigm). They sampled multiple species from a handful of sites in both Pennsylvania and Nevada. All of the samples are of the global panzootic lineage (GPL), and the authors are able to divide the genotypes into GPL-1, GPL-2 or “unassigned”. Unassigned genotypes are either a coinfection by both a GPL-1 & GPL-2 or a hybrid genotype. A major finding is that genotypes associated with bullfrogs are more often GPL-1, and genotypes from green frogs are more often “unassigned”. This leads to the conclusion that there is some level of specificity between host and pathogen genotypes and possibly coevolution.

Overall this paper is technically sound and presents a lot of new data. It also is a nice investigation of population structure and may change the way people think about the importance and role of Bd genotype in local disease dynamics. Most of my comments are about clarifications and interpretations.

Suggestions:

1. I think it is worth explaining what the basis for the GPL-1 vs. GPL-2 split is in the introduction and possibly discussion sections. GPL is really a clonal lineage that has fixed heterozygosity of some markers. GPL-1 is a version that has not had some consistent loss of heterozygosity (LOH) events that define GPL-2. A lot of these differences can be obscured by independent LOH events, and there may be intermediate genotypes that have only had some but not all of the LOH events. This paper is interesting in that it clearly picks up the two major groups and the intermediate. Ultimately it will be really fascinating if the unassigned genotypes in this study are sexually produced hybrids of these two sublineages. One thing worth noting is that European genotypes have largely been ignored in many pop gen studies, and I think it is important to consider that Europe may actually be the original location from which GPL dispersed across the world. See the phylogeny in Rosenblum 2013 which shows European strains at the base of the GPL clade.

We added an additional sentence explaining the original genomic characterizations of GPL-1 and GPL-2 in the introduction (Lines 91-95). Additionally, we added a discussion of Europe as an important site of high GPL genetic diversity (Lines 500-504).

2. There are a few places where terms like “potential ancestral population of GPL” are used, such as line 36 of the abstract. O’Hanlon and Byrne have shown that Asia has the highest amount of lineage diversity and is the most likely source of the GPL for that reason. At this point there is no evidence of any genotypes other than GPL in North America, but it is unclear when GPL might have colonized the region. Perhaps the authors are thinking about the source population of the MRCA of the known GPL strains in the world. As stated in point 1, I think we don’t know but I also think that you have to carefully consider European samples and collect more GPL genotypes from Asia before concluding that the GPL MRCA was North American.

We thank this reviewer for pointing out that Europe may also be an “ancestral population” for the Bd-GPL. We added a note to our discussion including this important possibility. (Lines 500-504). Additionally, we changed some of the wording in our abstract (Lines 48,58) and discussion (Lines 484, 486, 490, 502, 566) to better describe what we mean by “ancestral population of Bd-GPL”, which is “the source population of the most recent common ancestor of modern Bd-GPL lineages.” 

3. I mentioned fixed heterozygosity in point 1. Do the fluidigm markers allow heterozygosity estimation? If so, it would be interesting to see average observed heterozygosity across GPL-1, GPL-2 and unassigned. If the markers allow phasing and haplotype estimation shouldn’t it be possible that some of the markers in the unassigned would show more than 2 haplotypes under coinfection and never more than 2 for the hybrid hypothesis? I realize the authors do mention parasexual reproduction in the Discussion, and that could lead to more than 2 alleles at an individual locus. I think it’s fine to mention parasexual reproduction, but it is perhaps worth mentioning that the hybrid genotypes reported in Schloegel and Jenkinson have never had more than 2 alleles per locus.

We thank the reviewer for this suggestion. We now include a supplemental figure showing observed heterozygosity per individual for Bd-GPL1, Bd-GPL2 and the unassigned genotypes (Figure S6). We also now mention the fact that previously known hybrids have never had more than two alleles to the discussion (Line 544-545). In our data we find that even the isolates sometimes have more than 2 alleles at certain loci, perhaps due to copy number variation. We are currently working on applying new analytical methods to try to better address this question, but do not feel confident discriminating between coinfections and hybrids for this study. Future work isolating Bd at the PA site is ongoing and we hope to know more about the frequency of hybrid lineages at these sites in the near future. 

4. Line 47, some wording seems weird to me. Increased virulence will never be advantageous for a host. I think it may be more appropriate to say that strong host-specificity should favor intermediate virulence for that host.

Corrected as suggested. We edited this sentence for clarity (Lines 71-72).

5. Line 69, delete “remains untested”

Corrected as suggested (Line 100).

6. Line 86, what kind of gloves?

Added “nitrile” to this description (Line 136).

7. Line 141, what is “(N=13; 13,25)”

Edited this for clarity. Now it just includes the in-text citation and not the sample size. (Line 200)

8. Line 198-199, I think this should be clarified because as written it’s not clear why GPL-2 isolates would be heterozygous.

We edited this sentence for clarity (Line 271-272).

9. Line 247-248. Please explain what the test for within individual variation means. Variation is significantly structured by individuals?

The test for within-individual variation is included as a null hypothesis for the AMOVA. If most of the variation were to be attributed to within individual variation (rather than between sites or between species within a site), it would indicate a panmictic population. We included a sentence in our methods to explain this reasoning (Lines 244-247).

10. Figure S2, please explain what the X axis represents.

In Figure S2 the x-axis represents the component of covariance for the observed data (black line) and for the randomly permuted sample matrices (gray bars). We added this detail the figure legend for Figure S4 (previously Figure S2) (Line 1379) and labeled the axis on the figure itself. 

11. The associations in the PA population are interesting. These data really suggest future experimental studies. It might be worth expressing how you might test what is happening here with bullfrog GPL-1 association experimentally.

We agree that this is an exciting opportunity for future research. We added a paragraph to the discussion that mentions some potential future research approaches (Lines 468-483).

12. Line 382, suggest “possible” instead of “likely”. In most senses parasexual and sexual reproduction yield a pretty similar outcome. It could be hard to distinguish the two without observing intermediates or complete genome sequences.

Corrected as suggested (Line 539).

13. Line 386-387. This sentence was hard for me to follow. The use of parasexuality in nature is to present complex gene sets?

We edited this sentence for clarity (Line 543-545).

Reviewer #2: 

This study describes the genetic variation of a batrachochytrid fungus lineage (Bd-GPL) across several amphibian hosts in two regions of North America to explore pathogen diversification concerning host species. While the manuscript tackles an important topic, is well written, and provides detailed methods, in my opinion, the results are a little bit misleading, not providing enough support for the authors’ conclusions.

My main concern is that the sub-lineage category is ignored in the AMOVA analyses. Sub-lineage information might affect and confuse the relative contribution of host species to the genetic variance. In the data reduction analysis, the authors find that the sub-lineage explains the highest variation among samples (58.4%, Figure 1C), but this information is ignored in the subsequent analyses. Also, the PCA analysis clusters together samples from the two studied locations (Nevada and Pennsylvania), which, in turn, could change the interpretation of the results. A phylogenetic analysis integrating both regions could also improve result interpretation.

We thank the reviewer for their thoughtful suggestions. In this study, we used an AMOVA as a complimentary analysis to the DAPC clustering analysis. Each of these analyses use the raw genetic data and test for structure either based solely on the genetic patterns (as in the DAPC) or based on predefined, ecological categories (as in the AMOVA). The sub-lineage information is not included as a category in the AMOVA because we know for a fact that the variance in the molecular data is related to these sub-lineage categories (as the sub-lineages were calculated from the genetic data to describe the clusters within). The purpose of the AMOVA is to test our hypotheses that the variation in the genetic data is structured by either sampling locality or host species. While Figure 1 includes a PCA containing samples from both PA and NV, we note that Figures 2 and 3 include PCAs calculated from each separate sample group. We agree with the suggestion of including a combined phylogeny for all samples used in this study. We now include this figure in the supplement (Figure S3) and added a note in the methods to reflect this addition (Line 197-198).

Likewise, when describing the phylogenetic clades would be important to better characterize them. For instance, genetic diversity statistics (e.g., nucleotide diversity) could be computed between each sub-lineage to compare regions. Moreover, the coevolutionary hypothesis should be supported with further analysis of co-phylogeny congruence between pathogen and host.

We added a new supplementary figure (Figure S6) and related that compares heterozygosity among the different sub-lineages described in the paper. We feel that our general hypotheses of coevolution are ripe for further studies, including cophylogeny analyses, but that this analysis is outside the scope of this particular study. We added a new paragraph to the discussion that mentions some potential future research approaches, including cophylogenetic studies (Lines 468-483).

Another of my concerns is that Pennsylvania sampling sites could be dependent due to the geographic proximity. I recommend exploring the correlation between genetic and geographic distances (Mantel test).

We thank the reviewer for the recommendation and now include a mantel test (Figure S5) to explore the correlation of genetic and geographic distance for our samples. We found that there is a statistically significant, yet weak positive correlation between genetic and geographic distance for Pennsylvania. This is now mentioned in the paper (Line 350-352).

Finally, I think that one overlooked difference between Pennsylvania and Nevada sites that affects the pathogen diversification is the amphibian community structure. It would be worthwhile to characterize and integrate amphibian community metrics (e.g., richness per site) in this study.

We added more information about the amphibian species richness to the site descriptions (Lines 104-106, 108-109). We think this is an important point and added text to discuss the possible impacts that amphibian community diversity may have on Bd (Lines 462-464).

---

## [Editor Report · Decision Letter 1]

2 Mar 2022

Host species is linked to pathogen genotype for the amphibian chytrid fungus (Batrachochytrium dendrobatidis)

PONE-D-21-37079R1

Dear Dr. Byrne,

We’re pleased to inform you that your manuscript has been judged scientifically suitable for publication and will be formally accepted for publication once it meets all outstanding technical requirements.

Kind regards,

Louise A. Rollins-Smith

Academic Editor

PLOS ONE

I have looked carefully at your response to reviewers and your revised manuscript and figures.  It is my view that you have done a very good job of addressing the reviewer comments.  Thus, I am happy to recommend acceptance of the manuscript.

---

## [Editor Report · Acceptance letter]

4 Mar 2022

PONE-D-21-37079R1 

Host species is linked to pathogen genotype for the amphibian chytrid fungus (*Batrachochytrium dendrobatidis*) 

Dear Dr. Byrne:

I'm pleased to inform you that your manuscript has been deemed suitable for publication in PLOS ONE. Congratulations! Your manuscript is now with our production department. 

Kind regards, 

on behalf of

Dr. Louise A. Rollins-Smith 

Academic Editor

PLOS ONE